# Negative Impact of Elevated DNA Fragmentation and Human Papillomavirus (HPV) Presence in Sperm on the Outcome of Intra-Uterine Insemination (IUI)

**DOI:** 10.3390/jcm10040717

**Published:** 2021-02-11

**Authors:** Christophe Depuydt, Gilbert Donders, Ludo Verstraete, Johan Beert, Geert Salembier, Eugene Bosmans, Nathalie Dhont, Carmen Kerkhofs, Willem Ombelet

**Affiliations:** 1Department of Hormonology and Reproductive Health, AML, Sonic Healthcare, 2020 Antwerp, Belgium; christophe.depuydt@aml-lab.be (C.D.); ludo.verstraete@aml-lab.be (L.V.); johan.beert@aml-lab.be (J.B.); Eugene.bosmans@aml-lab.be (E.B.); 2Intermediate Structure for Human Body Material, AML, Sonic Healthcare, 2020 Antwerp, Belgium; 3Femicare, Clinical Research for Women, 3300 Tienen, Belgium; 4University Hospital Antwerpen, 2650 Antwerp, Belgium; 5Department of Obstetrics and Gynecology, Regional Hospital Heilig Hart, 3300 Tienen, Belgium; 6Department of Clinical and Molecular Pathology, AML, Sonic Healthcare, 2020 Antwerp, Belgium; geert.salembier@aml-lab.be; 7Genk Institute for Fertility Technology, ZOL Hospitals, 3600 Genk, Belgium; Nathalie.Dhont@zol.be (N.D.); Carmen.kerkhofs@zol.be (C.K.); Willem.Ombelet@zol.be (W.O.); 8Faculty of Medicine and Life Sciences, Hasselt University, 3500 Hasselt, Belgium

**Keywords:** quantitative real-time PCR, DFI, HDS, semen analysis, sperm chromatin structure assay (SCSA)

## Abstract

We wanted to determine the sperm DNA fragmentation index (DFI) cutoff for clinical pregnancies in women receiving intra-uterine insemination (IUI) with this sperm and to assess the contribution of Human Papillomavirus (HPV) infection on sperm DNA damage and its impact on clinical pregnancies. Prospective non-interventional multi-center study with 161 infertile couples going through 209 cycles of IUI in hospital fertility centers in Flanders, Belgium. Measurement of DFI and HPV DNA with type specific quantitative PCRs (HPV 6, 11, 16, 18, 31, 33, 35, 39, 45, 51, 52, 53, 56, 58, 59, 66 and 68) in sperm before its use in IUI. Clinical pregnancy (CP) rate was used as the outcome to analyze the impact on fertility outcome and to calculated the clinical cutoff value for DFI. A DFI criterion value of 26% was obtained by receiver operating characteristic (ROC) curve analysis. Couples with a male DFI > 26% had significantly less CPs than couples with DFI below 26% (OR 0.0326; 95% CI 0.0019 to 0.5400; *p* = 0.017). In sperm, HPV prevalence was 14.8%/IUI cycle. Sperm samples containing HPV had a significantly higher DFI compared to HPV negative sperm samples (29.8% vs. 20.9%; *p* = 0.011). When HPV-virions were present in sperm, no clinical pregnancies were observed. More than 1 in 5 of samples with normal semen parameters (17/78; 21.8%) had an elevated DFI or was HPV positive. Sperm DFI is a robust predictor of clinical pregnancies in women receiving IUI with this sperm. When DFI exceeds 26%, clinical pregnancies are less likely and in vitro fertilization techniques should be considered.

## 1. Introduction

Diagnosis of male infertility has mainly been based on the traditional semen parameters concentration, motility and morphology. It has, however, become increasingly clear that these standard sperm parameters recommended by the World Health Organization [1] are insufficient for the prognostic assessment of male fertility and for the evaluation of the fertility potential of a couple [2]. Among couples unable to conceive, infertility is to some extent attributable to a male factor in approximately 50% of cases [1,3,4]. While occupation, environmental and genetic factors are recognized as important causes of male infertility for some time [3], virologic threats by human papillomavirus (HPV) infections have only recently been described [5,6,7]. Indeed, although HPV infection is one of the most common viral infections worldwide, it is almost exclusively linked with pathogenesis of cancer. More recent studies have shown that male and couple subfertility may results from a HPV virion producing infection in one or both of the partners [6,7,8]. Also, the average lifetime probability of acquiring HPV among those with at least 1 opposite sex partner is more than 84.6% for women and 91.3% for men and more than 80% of women and men acquire HPV by age 45 years [9]. Therefore, we wonder whether HPV infection of sperm could be linked to the so-called ‘unexplained male infertility’, with normal semen parameters.

The pathogenesis of HPV infection in the cervix can be divided in two pathways, a non-infectious, cell transforming, cancer-inducing pathway [10,11] and an infectious, virion-producing pathway [12]. In sperm, the HPV DNA always originates from infectious virions only and is limited in time [8]. This is because HPV virions can only be produced in non-dividing cells which have a limited life span of 4 weeks, so virion production and shedding from desquamated cells once they reach the top of the epithelium are typically limited in time [8]. HPV virions can bind the syndecan-1 receptor at two distinct sites along the equator of the spermatozoon’s head [13,14,15]. This can cause detrimental effects on sperm parameters [14,16,17,18], damage to the sperm DNA [19] and impact on gamete interaction causing temporal subfertility [20]. In a previous prospective non-interventional multicenter study, we showed that when more than 66 HPV virions are present per 100 spermatozoa in the original sperm sample (the HPV virion per spermatozoon ratio cut off) no pregnancies occurred with intra-uterine insemination (IUI) [8].

The association between DNA damage and diminished reproductive outcomes has led to the introduction of sperm DNA integrity testing into the clinical assessment of male infertility [21]. Although a lot of people have been working with sperm DNA fragmentation, the field has been considered controversial, and implementation of the techniques in the clinic has been very slow [22]. The sperm chromatin structure assay (SCSA) is a flow cytometric technique that was first described by Evenson et al. [23] more than forty years ago, measuring the proportion of spermatozoa with impaired DNA integrity, expressed numerically as DNA fragmentation index (DFI) [24]. Studies have shown that the SCSA is an independent marker of fertility in vivo [25,26,27] or for success after intrauterine insemination (IUI) [28,29]. Although the SCSA test has been found to have the most stable clinical threshold values in relation to fertility, the clinical cutoff of DFI in semen that results in reduced clinical pregnancy rates varies from 25% to 30% [30].

In this prospective multi-center study, we measured both the DNA fragmentation index (DFI) with the sperm chromatin structure assay (SCSA) and HPV DNA with type specific quantitative PCRs in sperm before its use in IUI. Biochemical and clinical pregnancy rates were used as outcome parameters to analyze the impact of these parameters on fertility outcome and to determine the clinical cutoff value for DFI.

## 2. Materials and Methods

### 2.1. Study Population

The study was based on a cohort of 161 consecutive subfertile couples of a prospective non-interventional multi-center study between November 2017 and November 2018. All patients were treated with IUI at the Genk Institute for Fertility Technology, ZOL Hospitals, Genk, Belgium and in the Algemeen Medisch Laboratorium (AML) Intermediate structure for human body material for inpatient obstetrics and gynecology centers, Sonic Healthcare, Antwerp, Belgium. From the 161 subfertile couples (209 cycles), 152 women were inseminated with partner sperm (190 cycles) and 9 women with donor sperm (19 cycles). Consent was obtained to use the sperm rest fractions for DNA fragmentation and HPV testing. The study protocol was reviewed by the Institutional Review Board of the University hospital of Antwerp (Belgian registration number: B300201733597, approved on 9 October 2017). All participants signed consent that their rest material could be used for research purposes.

### 2.2. Patient Selection 

Included couples failed to conceive for at least 12 months. All patients underwent an infertility work-up prior to IUI treatment, as previously described [8]. For each male partner, at least two sperm samples were examined using the WHO guidelines [31]. Human immunodeficiency virus (HIV), hepatitis B/C virus and *Treponema pallidum* (syphilis) were measured in all couples as prescribed by Belgian law before receiving any treatment. As part of the infertility workup, we also always tested for the presence of *Trichomonas vaginalis*, *Chlamydia trachomatis* and *Neisseria gonorrhea* by PCR [32]. Only couples who were negative for all tested infectious diseases were included. Couples suffering from unexplained infertility, such as moderate male factor infertility, mild endometriosis and oligo-/anovulation, with at least one patent fallopian tube and a post-wash inseminating motile count (IMC) of >1 million were eligible for IUI, as previously described [33].

### 2.3. Semen Analysis and Capacitation 

Sperm samples were analyzed using the World Health Organization guidelines for semen analysis within one hour of production [31]. Semen samples were collected by masturbation on the day of insemination. Two to five days of sexual abstinence was asked of patients before sperm samples were collected. One of the previously described capacitation methods was used [8], according to the amount of progressive motile spermatozoa in the ejaculate (PMSC) and the Cytomegalovirus (CMV) status of both partners: swim-up (Sperm Preparation Medium, Origio, Denmark), density gradient (PureSperm^®^ 40/80, Nidacon, Sweden) or capacitation by removing the seminal plasma by washing with sperm preparation medium. Sperm antibodies were detected using the spermMar test kit for IgG and IgA (FertiPro, Beernem, Belgium). All rest sperm fractions of capacitations, instead of being discarded, were stored and transported at 4 °C to the laboratory (AML, Sonic Healthcare, Antwerp, Belgium). Twice a week, a DNA extraction was performed (M2000SP, Abbott Molecular Inc, Des Plaines, IL, USA).

### 2.4. Detection of HPV DNA by Type-Specific Real-Time Quantitative PCR (qPCR) Analysis in Sperm

HPV DNA testing was performed as described earlier [8]. We choose to measure HPV DNA in all sperm fractions that resulted from the capacitations (seminal plasma, swim-up fraction, pellet fraction, upper and lower gradient fractions and the rest fraction after washing the cell pellet) because this testing strategy gives less underestimation of the HPV prevalence (20%) compared to only measuring in 1/7 diluted sperm (40%) [8]. The extracted DNA was used in a quantitative real-time PCR assay that was clinically validated [34]. The assay can detect 18 HPV types: HPV6 E6, HPV11 E6, HPV16 E7, HPV18 E7, HPV31 E6, HPV33 E6, HPV35 E6, HPV39 E7, HPV45 E7, HPV51 E7, HPV52 E7, HPV53 E6, HPV56 E7, HPV58 E7, HPV59 E7, HPV66 E6, HPV67 L1 and HPV68 E7 [35]. HPV prevalence was defined as the detection of one or more of the above-mentioned HPV types. For high-risk HPV infection, one or more of the following 14 HPV types: HPV16, HPV18, HPV31, HPV33 HPV35, HPV39, HPV45, HPV51, HPV52, HPV56, HPV58, HPV59, HPV66 and HPV68 was detected. A **β**-globin quantitative real-time PCR was utilized to assess the quality of the extracted DNA and to calculate the number of cells [35]. The analytical sensitivity of the different HPV qPCRs ranges from 1 to 100 HPV copies/qPCR reaction [36]. To calculate the virions to spermatozoon ratio (HPV copies/spermatozoon), the number of HPV copies (virions) per ml sperm was divided by the number of spermatozoa per ml sperm. When in one sperm sample different HPV types were present, the number of HPV copies per ml of sperm for each individual HPV type was added and divided by the number of spermatozoa per ml. If in any one of the sperm fractions HPV DNA was detected, the sperm sample was considered HPV positive.

### 2.5. Sperm Chromatin Structure Assay (SCSA)

The flow cytometric SCSA, performed following the procedure, described previously [37], measures the percentage of sperm with fragmented DNA (DNA fragmentation index; DFI) as well as the percentage of sperm with high DNA stainability (HDS). The DFI% was calculated from the DFI frequency histogram obtained from the ratio between the red and total (red plus green) fluorescence intensity. DFI is the proportion of cells containing denatured DNA [37]. The HDS% was calculated based on the percentage of sperm with high levels of green fluorescence, which are thought to represent immature spermatozoa with incomplete chromatin condensation [37]. Briefly, on the day of capacitation, 100 µL of the original sperm sample that was used for IUI was immediately frozen and stored at −20 °C until SCSA was performed. On the day the SCSA was performed, the frozen sperm samples were quickly thawed (37 °C) and used immediately. Sperm cells were treated with a low pH detergent solution containing 0.1% Triton X-100, 0.15 M NaCl, and 0.08 N HCl (pH 1.2) for 30 s and then stained with an acridine orange solution (22.6 µM) at pH 6.0. For the flow cytometer setup and calibration, a reference sample was used as internal control sample. Samples were analyzed in batch and the internal control sample was analyzed in duplo at the beginning and end of each batch of samples. The intra-laboratory coefficient of variation (CV) of the internal control sample was 4.68% for DFI and 6.83% for HDS respectively.

### 2.6. IUI Protocol

We previously described how the patients were prepared for IUI (natural cycle, ovarian stimulation) and how IUI procedures were performed [8]. Out of the 209 IUI cycles, there were 204 natural cycles, and for 5 cycles, women received Clomid 100 mg per day from day 3 to 7, IUI was performed at 24–36 h post-hCG injection. The capacitated motile spermatozoa were inserted up to the uterine fundus and expelled into the uterine cavity. After insemination, women were asked to remain in supine position for 20 min [38,39]. A biochemical pregnancy (BP; early spontaneous abortion and or miscarriage) was diagnosed by the detection of HCG in serum or urine, without development of a clinical pregnancy. A clinical pregnancy (CP) was diagnosed when an unequivocal fetal heartbeat was detected by ultrasonography of at least one fetus [40]. As required by the Belgian Federal Agency for Medicine and Health Products (FAMHP), all pregnancy outcomes of IUI cycles were registered in a pregnancy registry. There was no cost or additional risk for patients and the data of this multi-center study were anonymized. All patients were informed about the aim of the study and were enrolled only if they gave their written informed consent. The gynecologists performing the IUI treatments were blinded from the SCSA and HPV results as to not influence the IUI treatment course. An IUI treatment course comprised between 1 up to 3 consecutive IUI cycles.

### 2.7. Statistical Analysis

The results were analyzed with MedCalc Statistical Software (MedCalc Software, Ostend, Belgium) [41]. All IUI cycles were dichotomized based on DFI and or HPV positivity in raw semen. DFI% and HPV status results were used to construct receiver operating characteristic (ROC) curves to determine a criterion value for distinguishing couples likely to achieve a clinical pregnancy from those unlikely to achieve a positive result. The area under the curve, confidence intervals of the area, and coordinates of the curve were used to determine the sensitivity and specificity of different threshold values. ROC analysis module was used to determine the clinical DFI% cut-off. Probability (*p*) values of <0.05 were considered statistically significant.

## 3. Results

### 3.1. Semen Parameters, IUI and Pregnancy

One hundred and sixty-one consecutive infertile couples undergoing 209 IUI cycles were enrolled. Couples characteristics and data concerning capacitated sperm parameters are summarized in Table 1. A total of 29 pregnancies was observed, 2 biochemical pregnancies (2/209; 1.0%) and 27 clinical pregnancies (27/209; 12.9%). Altogether, out of sperm samples from 169 different men, 31 tested positive for HPV, resulting in an HPV prevalence of 14.8% per IUI cycle and a high-risk HPV prevalence of 9.6% per IUI cycle (20/209). In more than one third of IUI cycles, the semen sample had normal WHO parameters (concentration, progressive motility and morphology) (78/209; 37.3%). Almost one in five of these semen samples with normal semen parameters had an DFI% above 26% and 7.7% (6/78) was HPV positive. More than 1 in 5 of samples with normal semen parameters (17/78; 21.8%) had an elevated DFI or was HPV positive.

### 3.2. ROC Analysis and Clinical DFI% Cutoff

Comparing DFI of sperm resulting in clinical pregnancies with that of sperm used in IUI cycles not resulting in a clinical pregnancy showed that no pregnancies resulted when DFI was above 26% (100.0% sensitivity and 36.3% specificity, ROC analysis area under the curve 0.73, (95% confidence interval (CI) 0.66 to 0.79; *p* < 0.0001). So a DFI cutoff of 26% was used to determine whether achieving a pregnancy would be very unlikely after IUI treatment (Figure 1).

The data of the different sperm parameters according to low (≤26%) or high (>26%) DFI are summarized in Table 1. A high DFI was found in 31.1% of IUI cycles, leading to a significantly lower chance to obtain a clinical pregnancy compared to the group with low DFI (OR 0.0326; 95% CI 0.0019 to 0.5400; *p* = 0.017). In the latter group, the mean DFI was 13.3%, compared to 41.9% in the former (*p* < 0.0001). 

The mean age of men with a low DFI was lower than of men with a high DFI (34.1 vs. 36.5 years; *p* = 0.026). Percentage of progressive sperm (43.5% vs. 34.2%; *p* = 0.0007) and normal sperm morphology (4.1% vs. 3.2%; *p* = 0.0069) was also significantly higher in the group with low DFI. 

There was no difference between the two DFI groups for female age, sperm concentration, progressive sperm concentration, IMC and HDS.

### 3.3. DFI vs. HPV Positivity in Sperm

The data of the different sperm parameters according to HPV positivity are summarized in Table 1. There was no difference in the age of men with HPV positive and HPV negative sperm. Compared to HPV negative men, HPV positive men had a significantly lower percentage progressive sperm count (*p* = 0.0058), lower IMC (*p* = 0.0086) and higher DFI (*p* = 0.0111) and HDS (*p* = 0.0025).

None of the 31 inseminations in which the sperm tested positive for HPV led to pregnancy, (100.0% sensitivity and 17.0% specificity, ROC analysis area under the curve 0.59, (95% CI 0.52 to 0.65 *p* > 0.05), even when DFI% was below 26% (Figure 1).

An overview of the detected HPV types is given in Table 2. The HPV types most frequently detected were HPV 6 (7/31; 22.6%), followed by HPV 67 (4/31; 17.6%) and HPV 16, 31, 53 and 66 each with 9.7% (3/31). To assess the impact of the different HPV types on sperm DNA damage, we classified single HPV type infections based on low (≤26%) or high (>26%) DFI and looked at the frequencies of occurrence of each HPV type in both groups (Table 2). HPV types 16, 45, 56 and 67 were almost exclusively detected in the group with DFI levels above 26%, whereas HPV 6, 39 and 52 were mainly detected in the group with DFI levels below 26%. The DFI% was not significantly higher in HR HPV positive sperm samples (31.4%) compared to sperm samples with LR HPV types (27.6%; *p* > 0.05).

To further assess the impact of the different HPV types on IUI outcome, we plotted the HPV virion per spermatozoon ratio in function of DFI (Figure 2.) The median HPV virion per spermatozoon ratio in HPV positive sperm was 0.46 (95% CI 0.08 to 1.28 HPV virions/spermatozoon). More than half of the HPV positive sperm samples (17/31; 54.8%) were below the 0.66 HPV virion per spermatozoon ratio cutoff (8), and 15/31 (48.4%) of sperm samples had a DFI below 26% cutoff. The majority of HPV positive sperm samples were above at least one or both cutoffs (26/31; 83.9%). Less than one fifth of HPV positive samples used for IUI (5/31; 16.1%) had a HPV virion per spermatozoon ratio below the 0.66 cutoff and a DFI below the 26% cutoff (quadrant with low DNA damage and more spermatozoa than HPV virions). In the quadrant with the double positives i.e., high DNA damage and more HPV virions than spermatozoa (above both cutoffs), the highest percentage of HR HPV types (3/4; 75%) was observed (Figure 2).

## 4. Discussion

With routine WHO sperm assessment, the precise etiology of male factor infertility remains undefined in 30–50% of patients [42,43]. Unlike unexplained male infertility with its normal semen parameters, idiopathic male infertility is diagnosed in the presence of altered semen characteristics without an identifiable cause and the absence of a female infertility cause [44]. In this study, we address two understudied tests that could explain part of the so called ‘unexplained’ male infertility: presence of HPV virions in sperm and DNA fragmentation test of spermatozoa.

In this prospective study, we determined the sperm DFI cutoff for clinical pregnancies to be 26%. The significantly lower success rates after using sperm with high DNA fragmentation are in agreement with Bungum et al. (2007) who reported a 10 times increased chance in achieving a clinical pregnancy when DFI was lower than 30% [29]. Additionally, Yang et al. (2011) using the SCSA, reported better pregnancy outcome after IUI when sperm with DFI lower than 25% was used [28]. In a prospective cohort study, Duran et al. (2002) showed that by using the TUNEL (terminal deoxynucleotidyl transferase-mediated dUDP nick-end labeling) technique a cut-off value of 12% was found above which no pregnancies occurred after IUI [45]. Not only measuring DNA fragmentation in raw and washed samples as related to other sperm preparations, but also the method used to measure DNA fragmentation (SCSA, TUNEL) will probably result in different cut-off values and a clinical validation of the method used predicting IUI outcome might be warranted [30].

We previously reported a four times lower clinical pregnancy rate per IUI cycle (2.9%) in women inseminated with HPV positive sperm compared to women who were inseminated with sperm that tested negative for HPV (11.1%/cycle; *p* = 0.0016) [8]. 

The overall (14.8%) and high-risk HPV (9.6%) prevalence in partner sperm per IUI cycle of the current study is comparable to the high-risk HPV prevalence rate in a previously reported study of women receiving IUI treatment [8]. This prevalence is also comparable to the 16% (confidence interval (CI): 10–23%) HPV prevalence in semen of seven studies focusing on infertile men that was reported in a recent meta-analysis and systematic review [46]. Because HPV prevalence varies from population to population, and in HPV positive men from ejaculate to ejaculate [47], it is important that HPV analysis is performed on the ejaculate used for IUI as was done in the current study. The explanations of these huge fluctuations in HPV on the sperm, probably is that HPV virions are not present on the sperm during passage through the genital tract but are catched upon release from the mucosa during ejaculation [48].

HPV infections exert their deleterious effects on fertility through its virions. In previous work, we calculated that the presence of any HPV infection with 0.66 HPV virions or more per spermatozoon did not lead to clinical pregnancy in women undergoing IUI [8,49], but that some sperm samples with a virion per spermatozoon ratio below the 0.66 cutoff could still result in clinical pregnancy. In the present study, we were able to demonstrate a strong association of the presence of HPV virions with increased DNA fragmentation of spermatozoa, leading to strongly reduced pregnancy rates after IUI cycles. However, in addition, in the current study, some of the HPV positive sperm samples used in IUI had low DFI rates and should have better odds. Plotting the DFI against the HPV/spermatozoon ratio reveals 2 different ways in which HPV virions exert their detrimental effect on fertility [20]. In the direct way, the virions bind the spermatozoa through the syndecan-1 [13,14,15] via their L1 protein, which in turn inhibits Aquaporin-8 functionality [50] and make spermatozoa more sensitive to oxidative stress, damaging the sperms DNA which results in elevated DFI and this both for HR and LR HPV types. As confirmed in our study, this could be HPV type specific since not all HPV types cause the same amount of DNA damage [51,52]. If HPV virions are catched up by spermatozoa during ejaculation [48], it could explain why some studies do not find a relationship between seminal HPV and DFI [53,54]. In our series, when HR HPV was present, there was a non-significant trend of higher DNA damage in spermatozoa compared to LR types. However, for fertility pathogenesis, the classification into HR and LR HPV types will probably not hold because some HR types are associated with higher DNA fragmentation (HPV 16, 18, 45) than others (HPV 52), whereas some LR HPV types such as HPV 6 and 67 were also found to cause high DNA damage. Because in the current study there is an underestimation of the HPV prevalence measured in the 18 different HPV types due to inhibitors present in sperm samples [8], and because other LR HPV types can be present in sperm we did not test for might also influence fertility, the reported HPV prevalence probably reflects an underestimation of the impact of HPV infections on fertility. Anyway, the link between HPV genotype and HPV risk type and DFI should be further explored in a larger study including more HPV isolations of each type in order to reach sufficient statistical power to draw valid conclusions on this matter.

A second explanation by which virions exert a negative effect on fertility is that infected spermatozoa can cause arrest in dividing embryonic cells by transferring HPV virions into blastocysts and oocytes [55]. Here, no DNA fragmentation of spermatozoa is involved, but direct effects of the invaded virions on the dividing embryonic cells. Once the HPV virions are present in the embryo, some HPV types may be more keen to damage the embryo. For strong carcinogenic types such as HPV 16, arrest of embryo division could be almost instantly [56].

Finally, since HPV virions can also be produced locally in the infected cervical cells of the partner, these can also bind to spermatozoa on their way to the oocyte. So, as long as active virions are produced by either partner, the odds to get pregnant are jeopardized, and as a consequence, it is only when the infectious HPV lifecycle with production of virions ends in both partners, that the chances for pregnancy normalize. As HPV virion production in non-dividing cells is limited in time [8], and the life cycle of spermatozoa is around 72 days, the estimated HPV virion induced damage to sperm DNA is probably shorter than the 7.5 months median duration HPV infections in men lasts [57]. We hypothesize that this can partly explain why infertile couples sometimes go through enduring, extensive infertility treatments such as IUI, in vitro fertilization (IVF) and intracytoplasmic sperm injection (ICSI) without success, and still spontaneously get pregnant months after having stopped all treatment efforts.

In 81 of the 209 IUI cycles (38.8%) with either the DFI > 26% and or HPV present, the pregnancy rate can be expected to be very low or even zero according to our results. In those cases, IUI should be avoided considering the price and burden associated with IUI which will differ substantially between countries. Studies to examine IVF and ICSI results in this specific DFI and/or HPV positive population are urgently needed to perform a good economic analysis considering the impact of DFI and HPV in artificial reproductive techniques (ART) programs. By including sperm DNA fragmentation analysis as a parameter in the routine diagnosis of male fertility patients in combination with HPV detection, it would be possible to avoid or to postpone unnecessary and unsuccessful IUI treatments, which would prevent a large burden of physical, financial and psychological stress caused by unsuccessful IUI trials. Although detection of infectious HPV virions could explain a substantial part of subfertility in our cohort, the elevated DFI is multifactorial and not exclusive to HPV infection, and can also be caused by other factors like oxidative stress [58]. Importantly, HPV infection in sperm is temporal, leading to DFI normalization after 3–6 months, and higher success rates can be achieved by postponing IUI until HPV negativation. Additionally, it can be advocated to screen female partners for HPV infections and try to achieve HPV clearance (e.g., by removal of the infected cervical tissues, vaccination, antiviral therapy, etc.), since pregnancy rates are higher in treated HPV positive women than in untreated women [59]. Our data support promoting prophylactic HPV vaccination to male subjects, as a recent study confirmed this ameliorates reproductive outcome of infertile couples with HPV semen infection beside lowering the risk for cancer [60]. Finally, we advocate not only to test male partners or sperm donors of women starting assisted reproductive techniques for HPV [8], but also to perform DFI [61].

## Figures and Tables

**Figure 1 jcm-10-00717-f001:**
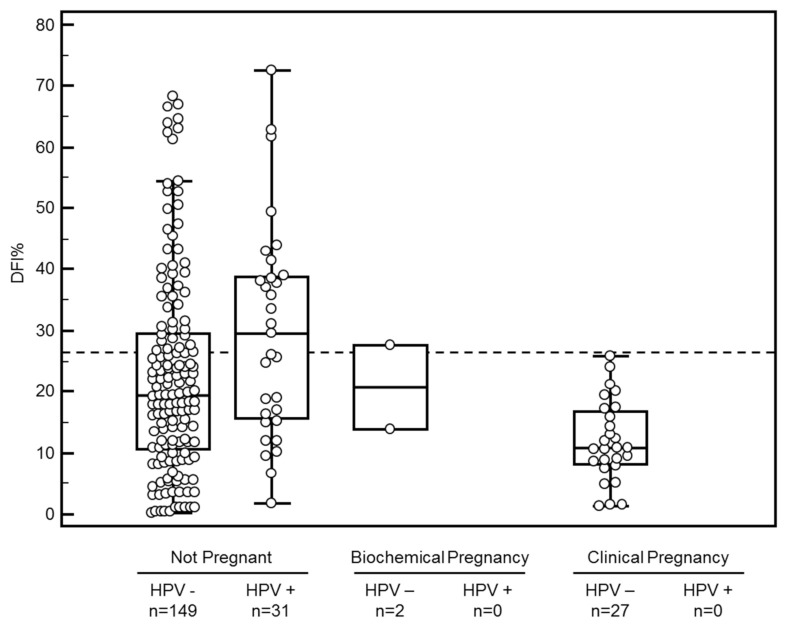
DFI% versus pregnancy outcome and HPV status. The dashed line represents the calculated DFI cutoff of 26% above which no clinical pregnancies could be achieved (sensitivity of 100% and specificity of 36.3%). Real time qPCR was used to quantify following HPV types: HPV 6, 11, 16, 18, 31, 33, 35, 39, 45, 51, 52, 53, 56, 58, 59, 67 and 68.

**Figure 2 jcm-10-00717-f002:**
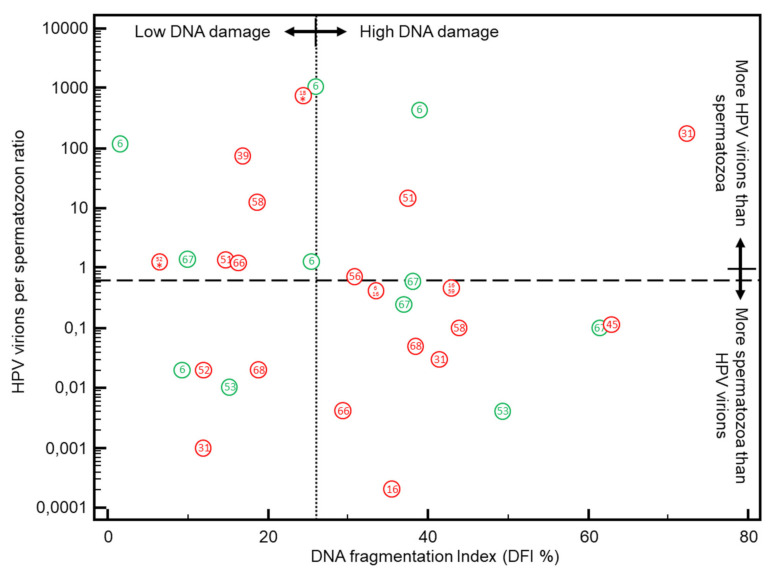
DNA fragmentation index (DFI%) versus HPV virions/spermatozoon ratio in neat sperm used for IUI. Dotted line = DFI cutoff (26%) above which no CP were observed with IUI (current study); Dashed line = the HPV virion per spermatozoon ratio cutoff (0.66) above which no CP were achieved with IUI (8); Green circles = low risk HPV types, Red circles = high risk HPV types. The number in the circle denotes the HPV type. 18* = multiple infection with HPV types 18, 6 and 59; 52* = multiple infection with HPV types 52, 53, 56 and 66. HPV types 6, 11, 16, 18, 31, 33, 35, 39, 45, 51, 52, 53, 56, 58, 59, 67 and 68 were quantified with real-time qPCRs.

**Table 1 jcm-10-00717-t001:** Couple characteristics and demographic data on 209 intra-uterine insemination (IUI) cycles divided according to low DNA fragmentation index (DFI) (≤26%) and high DFI (>26%) and Human Papillomavirus (HPV) status.

	All		DFI ≤ 26%		DFI > 26%		*p*-Value	HPV-Negative		HPV-Positive		*p*-Value
IUI Cycles Included (*n*; %)	209	100	144	68.9	65	31.1		178	85.2	31	14.8	
Female Age (mean; 95% CI)(Years)	32.3	31.5 to 33.1	32.3	31.3 to 33.2	32.3	30.9 to 33.7	NS	32.2	31.3 to 33.1	32.7	30.7 to 34.7	NS
Male Age (mean; 95% CI)(Years)	34.9	34.0 to 35.9	34.1	33.1 to 35.2	36.5	34.5 to 38.4	0.0259	34.8	33.8 to 35.8	35.6	32.7 to 38.6	NS
Sperm Concentration (Mean; 95% CI) (10^6^/mL)	47.3	40.3 to 54.4	49.4	40.6 to 58.3	43.1	31.3 to 54.9	NS	49.4	41.9 to 56.8	35.9	14.9 to 56.9	NS
Progressive Sperm Concentration (Mean; 95% CI) (10^6^/mL)	58.3	48.1 to 68.4	61.8	49.2 to 74.4	51.0	33.4 to 68.5	NS	61.1	49.9 to 72.3	42.2	18.8 to 66.7	NS
Percentage Progressive Sperm (Mean; 95% CI) (%)	40.5	38.0 to 43.0	43.5	40.6 to 46.4	34.2	29.7 to 38.7	0.0007	42.2	39.6 to 44.7	30.9	23.4 to 38.3	0.0058
Sperm Morphology (Mean; 95% CI) (%)	3.8	3.5 to 4.2	4.1	3.7 to 4.6	3.2	2.6 to 3.7	0.0069	3.9	3.5 to 4.3	3.3	2.3 to 4.3	NS
IMC (mean; 95% CI)(10^6^/mL)	9.5	7.7 to 11.4	10.4	8.0 to 12.9	8.1	5.6 to 10.6	NS	10.6	8.6 to 12.7	2.8	1.1 to 4.3	0.0086
DFI (mean; 95% CI) (%)	22.2	20.0 to 24.4	13.3	12.1 to 14.6	41.9	38.7 to 45.1	<0.0001	20.9	18.6 to 23.2	29.8	23.4 to 36.2	0.0111
HDS (mean; 95% CI) (%)	3.6	3.2 to 4.0	3.4	2.9 to 3.9	3.8	3.2 to 4.4	NS	3.3	2.9 to 3.7	5.0	3.5 to 6.5	0.0025
Normal Semen Parameters (WHO 2010) (*n*) (%/IUI cycle)	78	37.3	64	82.1	17	17.9	<0.0001	72	92.3	6	7.7	<0.0001
Biochemical Pregnancies (*n*) (%/IUI cycle)	2	1.0	1	0.7	1	1.5	NS	2	1.1	0	0	NS
Clinical Pregnancies (*n*) (%/IUI cycle)	27	12.9	27	18.8	0	0	0.0004	27	15.2	0	0	0.0417

IUI: intra-uterine insemination; DFI: DNA fragmentation index; HDS: High DNA stainability; IMC: inseminating motile count; NS: not significant *p* > 0.05.

**Table 2 jcm-10-00717-t002:** Overview of the detected HPV types.

HPV Type	Single	Multiple	Total		DFI ≤ 26		DFI > 26	
	*n*	*n*	*n*	%	*n*	%	*n*	%
6	5	2	7	22.6	4	80	1	20
11	-	-	-		-	-	-	-
16	1	2	3	9.7	0	0	3	100
18	-	1	1	3.2	1	100	0	0
31	3	-	3	9.7	1	33	2	67
33	-	-	-	-	-	-	-	-
35	-	-	-	-	-	-	-	-
39	1	-	1	3.2	1	100	0	0
45	1	-	1	3.2	0	0	1	100
51	2	-	2	6.5	1	50	1	50
52	1	1	2	6.5	2	100	0	0
53	2	1	3	9.7	1	50	1	50
56	1	1	2	6.5	0	0	1	100
58	2	-	2	6.5	1	50	1	50
59	-	2	2	6.5	-	-	-	-
66	2	1	3	9.7	1	50	1	50
67	4	-	4	12.9	1	25	3	75
68	2	-	2	6.5	1	50	1	50

HPV positive = detection of one of the following HPV types: HPV 6, 11, 16, 18, 31, 33, 35, 39, 45, 51, 52, 53, 56, 58, 59, 66, 67 or 68.

## Data Availability

The data presented in this study are available on request from the corresponding author. The data are not publicly available due to privacy.

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
