# Peer review of "Negative Impact of Elevated DNA Fragmentation and Human Papillomavirus (HPV) Presence in Sperm on the Outcome of Intra-Uterine Insemination (IUI)"

_jcm, 2021, doi:10.3390/jcm10040717_

Round 1

Reviewer 1 Report

In this cohort study the possible effect of increased DNA fragmentation index (DFI) and presence of HPV on pregnancy rates in IUI were evaluated in 161 infertile couples going through 209 IUI cycles. HPV was found in 14.8% /IUI cycles, and no clinical pregnancies were observed when HPV was detected. Furthermore, no clinical pregnancy was observed when the DFI exceeds 26%. Finally, an association between presence of HPV and high DFI was found.

  It is absolutely relevant to perform more studies about the predictive value of DFI and HPV on the chance to obtain pregnancy in fertility treatment, although a limitation of the present study may be a modest patient material. Furthermore, it is absolutely needed to obtain more knowledge about the possible mechanisms explaining a possible association.

  Although a lot of people have been working with sperm DNA fragmentation, the field has been considered controversial, and implementation of the techniques in the clinic has been very slow (see Rex AS et al.: DNA fragmentation in spermatozoa: a historical review. Andrology 2017; 5: 622-30).

  HPV prevalences varies from population to population, and in HPV positive men from ejaculate to ejaculate (Kaspersen et al.: Identification of multiple HPV types on spermatozoa from human sperm donors. Public           Library of Science ONE 2011; 6: e18095). Therefore, it is important that HPV analysis is performed on the ejaculate used for IUI (as done in this study), if one should expect to see a possible association between presence of HPV and pregnancy rate. This should be underlined. The explanations of these huge fluctuations in HPV on the sperm, probably is that HPV is not present on the sperm during passage through the genial tract but is catched up from the mucosa during ejaculation (see Fedder et al.: Seminal Human Papillomavirus (HPV) originates from the body surface and is not a frequent etiological factor in azoospermia. Studies in azoospermic and proven fertile, healthy men. Andrologia 2019; 51: e13202). Such suggestions should be included in the discussion.

  In several studies a possible relationship between seminal HPV and DFI already has been analyzed and usually are negative (e.g. Kaspersen et al.: No increased sperm DNA fragmentation index in semen containing human papillomavirus or herpesvirus. Andrology 2013; 1: 361-4), which fits very well with the hypothesis, that HPV is catched up by the sperm during ejaculation. This hypothesis should also be enclosed in the discussion.

  Line 31: You write “infectious HPV virions”. How can you know that they are infectious?

  About the female partners: Would it be possible to mention in how big proportion of the females hormone treatment is given and to mention the distribution between clomiphene and FSH treatment? Some authors have suggested that the optimal DFI threshold may depend upon the stimulation protocol applied for the female partner.

  Table 2: Some patients may have more than one HPV type? Could there be a possible synergistic effect?

  Figure 2: Should “Defragmentation Index” be “DNA fragmentation index”?

  References: Ref. 55 in the text should be ref 54. Ref. 56 in the text should be ref. 55. Ref. 57 in the text should be ref. 56 a.s.o.

Author Response

Reviewer 1

In this cohort study the possible effect of increased DNA fragmentation index (DFI) and presence of HPV on pregnancy rates in IUI were evaluated in 161 infertile couples going through 209 IUI cycles. HPV was found in 14.8% /IUI cycles, and no clinical pregnancies were observed when HPV was detected. Furthermore, no clinical pregnancy was observed when the DFI exceeds 26%. Finally, an association between presence of HPV and high DFI was found.

  It is absolutely relevant to perform more studies about the predictive value of DFI and HPV on the chance to obtain pregnancy in fertility treatment, although a limitation of the present study may be a modest patient material. Furthermore, it is absolutely needed to obtain more knowledge about the possible mechanisms explaining a possible association.

  Although a lot of people have been working with sperm DNA fragmentation, the field has been considered controversial, and implementation of the techniques in the clinic has been very slow (see Rex AS et al.: DNA fragmentation in spermatozoa: a historical review. Andrology 2017; 5: 622-30).

We have added the reference of Rex AS et al in the introduction.

  HPV prevalences varies from population to population, and in HPV positive men from ejaculate to ejaculate (Kaspersen et al.: Identification of multiple HPV types on spermatozoa from human sperm donors. Public           Library of Science ONE 2011; 6: e18095). Therefore, it is important that HPV analysis is performed on the ejaculate used for IUI (as done in this study), if one should expect to see a possible association between presence of HPV and pregnancy rate. This should be underlined. The explanations of these huge fluctuations in HPV on the sperm, probably is that HPV is not present on the sperm during passage through the genial tract but is catched up from the mucosa during ejaculation (see Fedder et al.: Seminal Human Papillomavirus (HPV) originates from the body surface and is not a frequent etiological factor in azoospermia. Studies in azoospermic and proven fertile, healthy men. Andrologia 2019; 51: e13202). Such suggestions should be included in the discussion.

We have added the reference of Kaspersen et al 2011 and Fedder et al 2019 in the discussion (lines 321-326) along with the following text: ‘]. Because HPV prevalences varies from population to population, and in HPV positive men from ejaculate to ejaculate [47], it is important that HPV analysis is performed on the ejaculate used for IUI as was done in the current study. The explanations of these huge fluctuations in HPV on the sperm, probably is that HPV virions are not present on the sperm during passage through the genial tract but are catched upon release from the mucosa during ejaculation [48].’

  In several studies a possible relationship between seminal HPV and DFI already has been analyzed and usually are negative (e.g. Kaspersen et al.: No increased sperm DNA fragmentation index in semen containing human papillomavirus or herpesvirus. Andrology 2013; 1: 361-4), which fits very well with the hypothesis, that HPV is catched up by the sperm during ejaculation. This hypothesis should also be enclosed in the discussion.

We have added the reference of Kaspersen et al 2013 and Cortés-Gutiérrez et al 2017 in the discussion (lines 342-344), aling with the following text: ‘If HPV virions are catched up by spermatozoa during ejaculation [48], it could explain why some studies do not find a relationship between seminal HPV and DFI [53,54].’

 Line 31: You write “infectious HPV virions”. How can you know that they are infectious?

Yes, it is correct we do not know if the viral DNA that was detected is infectious, although that for some couples the same HPV type was detected in both partners. You would need to do other experiments to show that the detected viral DNA was infectious. We deleted the word infectious.

About the female partners: Would it be possible to mention in how big proportion of the females hormone treatment is given and to mention the distribution between clomiphene and FSH treatment?

From the 161 women in the study, most women (n=156) had IUI in a natural cycle (n=204) and only 5 women received clomid. None of the women received FSH treatment. This was added in the IUI protocol on p4, line175.

Some authors have suggested that the optimal DFI threshold may depend upon the stimulation protocol applied for the female partner.

Because the majority of women in the current study did not receive clomid or FSH treatment (97.6%) it is not possible to ascertain if the DFI threshold may depend upon the stimulation protocol applied for the female partner.

  Table 2: Some patients may have more than one HPV type? Could there be a possible synergistic effect?

Yes, it is possible that infection with multiple HPV types is synergistic, 4 patients had more than one HPV type (see figure 2). Two had a high HPV virion per spermatozoon ratio above the 0.66 cut-off but low DNA damage (DFI<26%) and the other two had a HPV virion per spermatozoon ratio just under the 0.66 cut-off but had high DNA damage (DFI%>26%).

  Figure 2: Should “Defragmentation Index” be “DNA fragmentation index”?

Yes, it should be DNA fragmentation index, we corrected it in figure 2.

  References: Ref. 55 in the text should be ref 54. Ref. 56 in the text should be ref. 55. Ref. 57 in the text should be ref. 56 a.s.o.

References were corrected.

The entire research crew appreciates the extensive efforts and expertise of this reviewer. We are very thankful for the positive criticisms and are positively certain the comments have greatly ameliorated the content and presentation of the manuscript. We are therefore very, very grateful for this tremendous job performed.

Reviewer 2 Report

This is an interesting study that follows up on a previous publication from the authors that demonstrated significantly decreased pregnancy rates in the presence of active male HPV infection. In the present work the authors examine the combination of high DNA fragmentation and HPV infection in a cohort of patients undergoing IUI.

While the study is interesting and follows up on the authors previous striking findings, there are a couple of concerns as follows:

  1. As written, the overall objective of the paper is somewhat unclear: The authors state that the objective is to determine the DFI cutoff at which clinical pregnancy occurs, however, as the authors state in the paper this has been completed in a number of previous publications. Instead, the majority of the manuscript comments on HPV infection and DFI and the association between them. Similarly, the authors have previously shown that when more than 66 HPV virions are present per 100 spermtozoa, no pregnancies occurred, thus is would be helpful to know what the authors are hoping to add to the scientific literature in this study. It would be helpful for the authors to clearly state the objective and hypothesis behind this study.
  2. If there are almost no pregnancies seen in patients with HPV in their sperm, what is the benefit of also performing DNA fragmentation testing?
  3. No information is included about the breakdown of fertility diagnoses among this patient population. It would be expected that this would have an impact on the success rate of IUI.

Minor comments/questions

  1. The authors state that they wondered whether HPV infection could be linked to unexplained male infertility (normal semen parameters), but that is not what appears to be studied in this paper. The mean morphology was below 4% and HPV positive sperm also had abnormal progressive motility, thus these men do not have normal semen parameters and would not fall into the unexplained infertility category.
  2. It would be interesting to know the incidence of High DNA fragmentation and/or + HPV in couples with truly unexplained infertility (normal female examination and normal SA) and this would address the question that the authors pose in both the introduction and conclusion?
  3. In the discussion, the authors state that they were able to demonstrate a strong association of the presence of HPV virions with increased DNA fragmentation of spermatozoa, leading to reduced pregnancy rates. They have previously published significantly decreased pregnancy rates when HPV is present in IUI samples, so how is this study adding something novel?
  4. Similarly, the authors state that some of the HPV samples in the study had low DFI and should have had better odds, however, if they have previously shown that simply the presence of HPV virions decreases pregnancy rates, why would they expect that?
  5. In the discussion the authors suggest removing infected cervical tissue from women with HPV infections as a treatment prior to fertility treatments: Is there a treatment the authors are suggesting other than something invasive such as LEEP or Cone to remove pre-cancerous HPV-infected cervical tissue? These procedures are both invasive with significant associated risks.

Author Response

Reviewer 2

This is an interesting study that follows up on a previous publication from the authors that demonstrated significantly decreased pregnancy rates in the presence of active male HPV infection. In the present work the authors examine the combination of high DNA fragmentation and HPV infection in a cohort of patients undergoing IUI.

While the study is interesting and follows up on the authors previous striking findings, there are a couple of concerns as follows:

As written, the overall objective of the paper is somewhat unclear: The authors state that the objective is to determine the DFI cutoff at which clinical pregnancy occurs, however, as the authors state in the paper this has been completed in a number of previous publications. Instead, the majority of the manuscript comments on HPV infection and DFI and the association between them. Similarly, the authors have previously shown that when more than 66 HPV virions are present per 100 spermatozoa, no pregnancies occurred, thus is would be helpful to know what the authors are hoping to add to the scientific literature in this study. It would be helpful for the authors to clearly state the objective and hypothesis behind this study.

The main objective of this study was to determine the DFI cutoff at which a clinical pregnancy occurs. The hypothesis behind this study is that DNA sperm damage can be used to predict clinical pregnancies, but DFI is multifactorial and we wanted to assess the contribution of HPV infection on sperm DNA damage and its impact on clinical pregnancies.

We clarified this in the abstract and added the data about semen samples with normal sperm parameters that had an elevated DFI and were HPV positive.

If there are almost no pregnancies seen in patients with HPV in their sperm, what is the benefit of also performing DNA fragmentation testing?

We previously observed that having normal sperm parameters does not guarantee clinical pregnancies. We assumed that semen samples with normal semen parameters would have normal DFI and not be infected with HPV.

Although HPV infection can explain in part why no clinical pregnancy occurs, more than one third of sperm samples used for IUI had normal WHO sperm parameters (concentration above 15 million/ml, more than 32% progressive spermatozoa and a normal morphology of 4% or more) (78/209 IUI cycles; 37.3%), but almost one in five of these sperm samples with normal WHO parameters (14/78; 17.9%) the DFI% was above 26%.

Therefore, it appears that performing DFI allows to detect a large group of patients with normal WHO criteria (concentration, progressive motility and morphology; 17.9%) that will not benefit from IUI.

Using HPV testing as a triage we now know that an additional 3.8% (3/78) of these patients with normal WHO semen samples parameters and DFI% below 26% will not get pregnant using IUI either.

Testing for HPV and not perform DFI would only identify 6 out of 78 with normal semen parameters who did not achieve a clinical pregnancy instead of 14 out of 78. Combining DFI and HPV resulted in identifying 17 cases (3 HPV + and DFI<26% + 14 with DFI>26%) out of 78 with normal sperm parameters who will not benefit from IUI.

No information is included about the breakdown of fertility diagnoses among this patient population. It would be expected that this would have an impact on the success rate of IUI.

Yes, we agree, we added the number of natural cycles (204/209) and cycles with ovarian stimulation (5/209) (p4; 2.6 IUI protocol) and how many sperm samples had normal WHO semen parameters (concentration, progressive motility and morphology; 37.3%) (Table 1).

Minor comments/questions

The authors state that they wondered whether HPV infection could be linked to unexplained male infertility (normal semen parameters), but that is not what appears to be studied in this paper. The mean morphology was below 4% and HPV positive sperm also had abnormal progressive motility, thus these men do not have normal semen parameters and would not fall into the unexplained infertility category.

We clarified this by adding in how many IUI cycles with normal WHO semen parameters high DFI above 26% and HPV positivity was detected in this group (see Table 1 and results). More than one in five of semen samples with normal semen parameters has a DFI above 26% (14/78; 17.9%) or was positive for HPV even when DFI was below 26% (3/64;4.7%).

It would be interesting to know the incidence of High DNA fragmentation and/or + HPV in couples with truly unexplained infertility (normal female examination and normal SA) and this would address the question that the authors pose in both the introduction and conclusion?

In the current study we did not systematically measure the HPV status in the female partners, and in the next study we did measure the HPV status in both partners (957 IUI cycles) but did not determined the DFI. For the 5 IUI cycles with low DNA damage and more spermatozoa than virions (lower left corner in figure 2) all female partners were HPV positive at the moment they received IUI. In the larger study without DFI measurement intermediate analysis showed reduced clinical pregnancies when either male, female or when both partners were HPV positive compared to when both partners were HPV negative. The HPV prevalence per IUI cycle was 25.8% (247 HPV+/957 IUI cycles).

In the discussion, the authors state that they were able to demonstrate a strong association of the presence of HPV virions with increased DNA fragmentation of spermatozoa, leading to reduced pregnancy rates. They have previously published significantly decreased pregnancy rates when HPV is present in IUI samples, so how is this study adding something novel?

HPV infection can cause reduced fertility directly by binding the syndecan 1 receptor present on spermatozoa and cause an increase in DFI (high DNA damage), but the virions can also be transferred into the oocyte and indirectly cause a reduction of fertility. In most of the sperm samples with DFI >26% (n=65) HPV was detected in 26.1% (n=17), and other causes (than an HPV infection) should be looked for. Detecting HPV in sperm is important because virion productive infections are always limited in time, which means that when the HPV infection has been cleared fertility could be restored.

Similarly, the authors state that some of the HPV samples in the study had low DFI and should have had better odds, however, if they have previously shown that simply the presence of HPV virions decreases pregnancy rates, why would they expect that?

It is the increase in DFI (sperm DNA damage) above 26% that impacts pregnancy rates the most (even when no HPV was detected). Even when the semen parameters are normal and the DFI is below 26% you will not get pregnant if HPV was detected. The HPV virions present in the female partner probably adds a negative effect on fertility outcome. We unfortunately did not test all women for HPV at the moment of IUI (only 161 of the 209 IUI cycles), but all the female partners of the 5 cases in the lower left corner of Figure 2 (low DNA damage and more spermatozoa than HPV virions) were HPV positive at time of IUI. You probably need to add the female HPV viral load into the equation (see previous point).

In the discussion the authors suggest removing infected cervical tissue from women with HPV infections as a treatment prior to fertility treatments: Is there a treatment the authors are suggesting other than something invasive such as LEEP or Cone to remove pre-cancerous HPV-infected cervical tissue? These procedures are both invasive with significant associated risks.

We agree there is other options, albeit less documented. We adapted the text (L389-392)as follows: ‘Also it can be advocated to screen female partners for HPV infections and try to achieve HPV clearance (e.g. by removal of the infected cervical tissues, vaccination, antiviral therapy….), since pregnancy rates are higher in treated HPV positive women than in untreated women [59].’

The entire research crew appreciates the extensive efforts and expertise of this reviewer. We are very thankful for the positive criticisms and are positively certain the comments have greatly ameliorated the content and presentation of the manuscript. We are therefore very, very grateful for this tremendous job performed.

Round 2

Reviewer 1 Report

Thank you. I have no further comments. 

Reviewer 2 Report

I believe all comments have been addressed.